

# Effect of body size on the oral pharmacokinetics of oxytetracycline in rainbow trout (*Oncorhynchus mykiss*)

Orhan Corum[1], Erdinc Turk[1], Duygu Durna Corum[1], Ertugrul Terzi[2], Mustafa Cellat[3], Önder Yıldırım[4] and Kamil Uney[5]

[1] Department of Pharmacology and Toxicology, Faculty of Veterinary Medicine, University of Hatay Mustafa Kemal, Hatay, Türkiye

[2] Department of Veterinary Medicine, Devrekani TOBB Vocational School, University of Kastamonu, Kastamonu, Türkiye

[3] Department of Physiology, Faculty of Veterinary Medicine, University of Hatay Mustafa Kemal, Hatay, Türkiye

[4] Fisheries Faculty, Mugla University, Mugla, Türkiye

[5] Department of Pharmacology and Toxicology, Faculty of Veterinary Medicine, University of Selcuk, Konya, Türkiye

Corresponding author
Önder Yıldırım,
onderyildirim@mu.edu.tr

## ABSTRACT

**Objective**. The aim of this study was to determine the plasma pharmacokinetics of oxytetracycline (OTC) in rainbow trout (*Oncorhynchus mykiss*) of different body sizes.
**Methods**. The research was carried out on three groups as small (30–50 g), medium (90–110 g) and large (185–215 g) body sizes at $8 \pm 0.5$ °C. OTC was administered orally at a dose of 60 mg/kg to all groups. Blood samples were taken at 19 different sampling times until the 384 h after oxytetracycline administration. The plasma concentrations of OTC were measured using high pressure liquid chromatography-ultraviolet and pharmacokinetic parameters were evaluated using non-compartmental analysis.
**Results**. OTC was detected in small-body sized fish until the 336 h and in medium and large-body sized fish until the 384 h. The elimination half-life of OTC was 85.46, 87.24 and 86.98 h in the small, medium and large body size groups, respectively. The peak plasma concentration increased from 0.66 to 1.11 µg/mL, and the area under the plasma concentration-versus time curve from zero (0) h to infinity ($\infty$) increased from 87.86 to 151.52 h*µg/mL, in tandem with the increase in fish body size. As fish body size increased, volume of distribution and total body clearance decreased.
**Conclusion**. These results show that the pharmacokinetics of OTC vary depending on fish size. Therefore, there is a need to reveal the pharmacodynamic activity of OTC in rainbow trout of different body sizes.

## INTRODUCTION

Fish consumption has increased as people change their dietary habits to healthy diet. Türkiye ranks second after Iran in rainbow trout, which is grown in 77 countries around the world (*Yıldırım & Cantas, 2022*). Trout is a fish species widely grown in inland waters

and constitutes 37.2% of the total fish production in our country (*Uney et al., 2021*; *Corum et al., 2023a*). Rainbow trout (*Oncorhynchus mykiss*) is of great economic importance due to its rapid growth, tolerance to relatively high temperatures and suitability for hatchery farming (*Stanković, Crivelli & Snoj, 2015*). Septicemia, furunculosis, vibriosis, enteric red mouth disease, and rainbow trout fry syndrome are common in rainbow trout and antibiotics are used in their treatment (*Verner-Jeffreys & Taylor, 2015*; *Terzi et al., 2020*).

Oxytetracycline (OTC) is a tetracycline group antibiotic that has been widely used in the treatment of bacterial fish diseases for many years. OTC has advantages such as having a wide spectrum of action, good tissue penetration, low cost and minimal toxic effects (*Pinto et al., 2023*). OTC exhibits a bacteriostatic action by inhibiting bacterial protein synthesis. It achieves this by binding to the 30S ribosomal subunit of bacteria and blocking mRNA binding of tRNA (*Leal, Santos & Esteves, 2019*). It is recommended to use OTC at a dose of 60-100 mg/kg for 5–10 days in the treatment of fish diseases such as furunculosis, vibriosis, septicemia, rainbow trout fry syndrome and enteric red mouth disease (*Verner-Jeffreys & Taylor, 2015*; *Leal, Santos & Esteves, 2019*; *Manna et al., 2022*).

With age, anatomical and physiological differences such as body water/fat ratio, plasma protein concentration, enzyme capacity, and maturation of organs occur in living things (*Coskun et al., 2023*). Among vertebrates, fish are one of the creatures whose body mass ratio changes the most throughout their life cycle (*Clark & Farrell, 2011*). Since fish are heterotherms, environmental factors, especially temperature, are very important in their growth and development. For this reason, it is more accurate to evaluate their vital periods in terms of size rather than age. In fish, body components (fat, water, protein and muscle), organ weights and metabolism vary depending on body size (*Denton & Yousef, 1976*; *Weatherley, 1990*; *Schultz & Hayton, 1994*).

Although many studies have been conducted on the pharmacokinetics of OTC in rainbow trout (*Li et al., 2015*; *Corum et al., 2023b*), no study has been found comparing its pharmacokinetics in fish of different body sizes. OTC is used on fish of all body sizes in case of bacterial disease (*Rigos & Smith, 2015*). Anatomical and physiological differences in fish due to body size differences may change the pharmacokinetics of drugs. This change also alters the pharmacodynamics of the antibiotic. As a result, treatment failure due to insufficient drug concentration or toxic effects due to excessive concentration may occur. This causes resistance development in bacteria (*Coskun et al., 2023*). The body distribution of some toxic substances in fish varies depending on body size (*Newman & Mitz, 1988*; *Tarr, Barron & Hayton, 1990*; *Schultz & Hayton, 1994*). However, to our knowledge, no study has been found that reveals the pharmacokinetic change of any drug, including OTC, depending on body size in fish. We hypothesized that the pharmacokinetics of OTC may vary depending on fish size. The aim of this study is to determine the pharmacokinetic change following oral administration of OTC at a dose of 60 mg/kg in fish of different body sizes.

## MATERIALS & METHODS

### Chemicals

The analytical standard of OTC hydrochloride ($\geq$95%) was obtained from Tokyo Chemical Industry (Tokyo, Japan). Acetonitrile was used in high-performance liquid chromatography (HPLC)-grade (VWR International, Fontenay-sous-Bois, France). Other chemicals were provided from Merck (Darmstadt, Germany). Oral commercial preparation of OTC (Oksifish 75% Medicated Premix, Medicavet, Istanbul/Türkiye) was used for drug administration to fish.

### Animals

The experiment on fish was carried out in a local fish farm (Kastamonu/Türkiye). The research was carried out on a total of 342 healthy rainbow trout with small ($n = 114$, 30–50 g), medium ($n = 114$, 90–110 g) and large ($n = 114$, 185–215 g) body sizes. Fish that had not taken any medication in the last two months and had no signs of disease or trauma were included in the study. Fish were kept in concrete ponds with a continuous flow of spring water (temperature: $8 \pm 0.5\,°C$, pH: $8.1 \pm 0.2$) under natural daily lighting conditions. The fish were taken into the ponds two weeks before the study to allow them to adapt to the environment. Fish were fed with drug-free commercial fish feed (Sibal, Sinop, Türkiye). To prevent the effect of food content on the absorption of OTC, fish were fasted for 12 h before and after drug administration. To reduce traumatic wounds and stress that may occur in trout, drug administration and blood collection were carried out under tricaine methanesulfonate (MS-222, 200 mg/L) anesthesia. There is no other practice that will cause pain and suffering in fish other than oral drug administration and blood collection. Additionally, six different fish were used at each sampling time to minimize the stress and pain the fish. No different analgesic was administered to prevent a possible effect on the pharmacokinetics of OTC. In the research, we took blood at a level that would not affect the fish's physiology, and then we allowed them to resume their normal lives. Therefore, no euthanasia procedure was used. After blood collection, the fish were taken to a different pool from the other fish and kept there. The experimental was approved (2021/23) by the Kastamonu University Animal Experiments Local Ethics Committee (Kastamonu, Türkiye).

### Experimental design

For drug administration to fish, the oral preparation of OTC was diluted with injection water at the concentration of 12 mg/mL for small body size and 30 mg/mL for medium and large body size. A total of 342 fish were randomly divided into three groups for small ($n = 114$), medium ($n = 114$) and large ($n = 114$) body size. Drug administration and blood collection were conducted under MS-222 (200 mg/L) anesthesia. OTC was administered orally through gastric gavage at a dose of 60 mg/kg to all three groups. Blood samples from small body size (0.4 mL) and other groups (one mL) were collected from the caudal vessel into heparin-containing anticoagulant tubes the following time points: at 0 (control), 0.25, 0.5, 1, 2, 4, 8, 12, 24, 48, 72, 96, 120, 144, 192, 240, 288, 336 and 384 h after of OTC

administration. Plasma obtained by centrifuging blood samples at 4000 g for 10 min was stored at −80 °C until analysis.

## Oxytetracycline analysis

OTC analysis from plasma samples was performed using a HPLC-UV according to previously reported methods (*Corum et al., 2023a*; *Corum et al., 2023b*). Briefly, 100 μL of plasma was transferred to two mL microcentrifuge tubes. Then, 200 μL buffer/EDTA (0.1 M disodium EDTA containing 0.1 M sodium phosphate) and 50 μL perchloric acid (60%) were added to the plasma. The mixture was vortexed for 45 s and then centrifuged at 15.000 g for 10 min. The supernatant was transferred to autosampler vials and 50 µl was injected into the HPLC system. HPLC system consists of a column oven (CTO-10A), a pump (LC-20AT), a degasser (DGU-20A), an auto-sampler (SIL 20A), and an UV–VIS detector (SPD-20A). Separation was carried out with an inertsil ODS-3 column (4.6 × 250 mm; 5 µm; GL Sciences, Japan) kept at 40 °C. The UV detection wavelength was set at 260 nm. The mobile phase consisted of 0.01 M trifluoroacetic acid and acetonitrile (80:20, v/v). The flow rate was 0.8 mL/min.

The chromatographic procedure was validated following the guidelines provided by the European Medicines Agency (*European Medicines Agency, 2011*). The stock solution of OTC was prepared in purified water to obtain a concentration of 1 mg/mL. Calibration standards (0.04–4 µg/mL) and quality control samples (0.1, 0.4 and 1 µg/mL) were prepared by adding working standard solutions (0.04–4 µg/mL) of OTC into blank fish plasma. The calibration curve of OTC was linear ($R^2$ >0.9990) between 0.04 and 4 µg/mL. For the purpose of determining recovery, precision, and accuracy, quality control samples of OTC at low (0.1 µg/mL), medium (0.4 µg/mL), and high (1 µg/mL) concentrations were utilized. The recovery of OTC ranged from 92% to 97%. The lower limit of quantification (LLOQ) was 0.04 µg/mL for OTC in rainbow trout plasma with the coefficient of variation less than 20% and the bias of ±15%. The intra-day and inter-day coefficients of variation were ≤6.2% and ≤7.6%, respectively. The intra-day and inter-day bias were ±6.1% and ±7.3%, respectively.

## Pharmacokinetic analysis

Plasma concentrations were analyzed by noncompartmental analysis using the WinNonlin 6.1.0.173 software. The concentrations of plasma are displayed as mean ± standard deviation. Pharmacokinetic parameters were calculated based on mean concentrations as in previous studies (*Corum et al., 2022*; *Durna Corum et al., 2022*). The peak plasma concentration ($C_{max}$), time to reach peak plasma concentration ($T_{max}$), volume of distribution ($V_{darea}/F$), terminal elimination half-life ($t_{1/2\lambda z}$), area under the plasma concentration *versus* time curve (AUC), AUC extrapolated from $t_{last}$ to$_\infty$ in % of the total AUC ($AUC_{extrap}$ %), total body clearance (CL/F) and mean residence time (MRT) were determined.

## RESULTS

The semi–logarithmic plasma concentration–time curves and pharmacokinetic parameters of OTC following oral administration of 60 mg/kg to different size of rainbow trout are

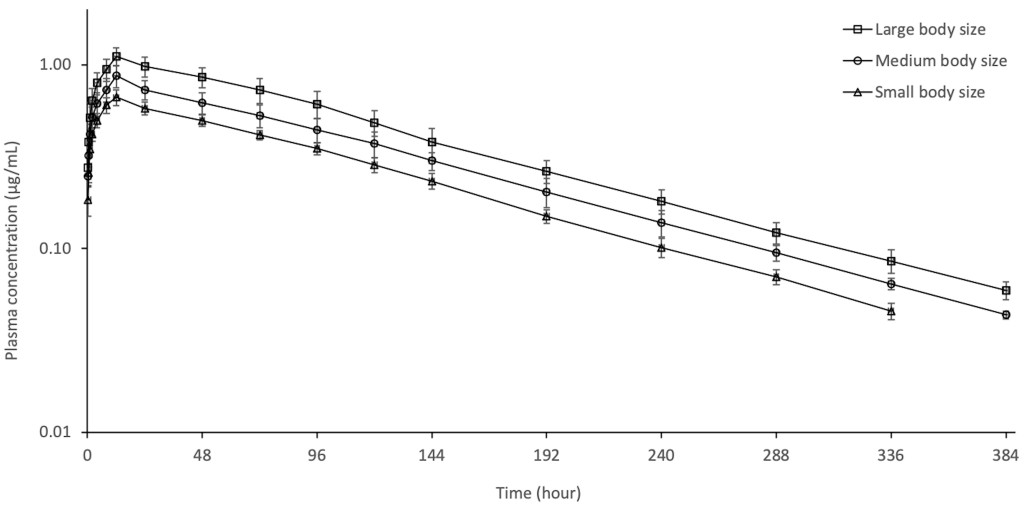

**Figure 1** Semi-logarithmic plasma concentration–time curves of oxytetracycline following oral administration of 60 mg/kg dose in rainbow trout of different body size at 8 ± 0.5 °C ($n = 6$).

**Table 1** Plasma pharmacokinetic parameters of oxytetracycline following oral administration of 60 mg/kg dose in rainbow trout of different body size at 8 ± 0.5 °C.

| Parameters | Small body size | Medium body size | Large body size |
|---|---|---|---|
| $t_{1/2\lambda z}$ (h) | 85.46 | 87.24 | 86.98 |
| $AUC_{0-last}$ (h*$\mu$g/mL) | 82.24 | 108.48 | 144.10 |
| $AUC_{0-\infty}$ (h*$\mu$g/mL) | 87.86 | 113.95 | 151.52 |
| $AUC_{extrap}$ (%) | 6.40 | 4.80 | 4.90 |
| $MRT_{0-\infty}$ (h) | 124.73 | 129.36 | 128.97 |
| CL/F (L/h/kg) | 0.68 | 0.53 | 0.40 |
| $V_{darea}$/F (L/kg) | 84.20 | 66.27 | 49.69 |
| $C_{max}$ ($\mu$g/mL) | 0.66 ± 0.07 | 0.87 ± 0.12 | 1.11 ± 0.12 |
| $T_{max}$ (h) | 12.00 | 12.00 | 12.00 |

**Notes.**

$t_{1/2\lambda z}$, elimination half-life; AUC, area under the concentration-versus time curve; $AUC_{extrap}$ %, area under the plasma concentration–time curve extrapolated from $t_{last}$ to ∞ in % of the total AUC; MRT, mean residence time; CL/F, total body clearance; $V_{darea}$/F, apparent volume of distribution; $C_{max}$, peak plasma concentration; $T_{max}$, time to reach the peak plasma concentration.

shown in Fig. 1 and Table 1, respectively. OTC was detected in plasma up to 336 h in small body size and up to 384 h in other fish body sizes. The $C_{max}$ value for fish in small, medium and large body sizes was 0.66 ± 0.07, 0.87 ± 0.12 and 1.11 ± 0.12, respectively. At the last sampling times, the concentration of OTC dropped to 0.05 ± 0.01, 0.04 ± 0.01 and 0.06 ± 0.01 in small, medium and large body size, respectively. OTC showed a long $t_{1/2\lambda z}$ (85.46–87.24 h) in small, medium and large body size. As fish body size increased, AUC and $C_{max}$ increased and CL/F and $V_{darea}$/F decreased. The $t_{1/2\lambda z}$ and $T_{max}$ were similar in all groups. The $AUC_{extrap}$ value for all group was less than 20%.

## DISCUSSION

OTC is one of the most widely used antibiotics in the treatment of bacterial infections in fish. There are many studies on the pharmacokinetics of OTC in fish (*Li et al., 2015*; *Corum et al., 2023b*). Although OTC is used in fish of all body sizes (age), no study has been found that demonstrates changes in pharmacokinetics depending on body size. In this study, the pharmacokinetic changes of OTC depending on small, medium and large body size were revealed for the first time. The pharmacokinetics of OTC were observed to vary significantly depending on body size in rainbow trout.

The $t_{1/2\lambda z}$ of OTC in small, medium and large body sized rainbow trout was similar. However, the $t_{1/2\lambda z}$ of trifluralin was prolonged from 15.5 h to 144 h depending on body size (*Schultz & Hayton, 1994*). The $t_{1/2\lambda z}$ of OTC after oral administration at different temperatures (5−16 °C), doses (50–150 mg/kg) and body size (246–558 g) in rainbow trout was reported as 23.20–479.43 h (*Björklund & Bylund, 1990*; *Rogstad et al., 1991*; *Uno, Aoki & Ueno, 1992*; *Abedini, Namdari & Law, 1998*). These results indicate that $t_{1/2\lambda z}$ of OTC is highly variable. Since fish are poikilotherm, the pharmacokinetics of OTC changed depending on temperature (*Björklund & Bylund, 1990*). The variability of $t_{1/2}\lambda_z$ in rainbow trout may be due to differences in temperature, oral administration method (feed or gavaj), drug formulation and analysis methods. $T_{1/2\lambda z}$ is a hybrid parameter depending on $Cl_T$ and $V_d$. In this study, the lack of change in $t_{1/2\lambda z}$ may be due to the decrease in $Cl_T$ and $V_d$ depending on the body size.

The $V_{darea}/F$ after oral administration in small, medium and large fish was 84.20, 66.27, and 49.69 L/kg, respectively. $V_{darea}$ was between 0.76 and 2.99 L/kg after intravenous administration in rainbow trout (*Black et al., 1991*; *Abedini, Namdari & Law, 1998*; *Corum et al., 2023b*). OTC has a wide $V_{darea}$ due to its lipophilic structure and partially low (52.2–55.3%) binding to plasma proteins (*Björklund & Bylund, 1991*; *Corum et al., 2023b*). The most appropriate way to determine $V_{darea}$ is intravenous administration, as bioavailability will have an impact on extravascular administration routes. The oral bioavailability of OTC in rainbow trout was quite low (0.6−5.6%, (*Uno et al., 1997*; *Rigos & Smith, 2015*)). The calculation of $V_{darea}/F$ was based on the equation $V_{darea}/F = dose/concentration$. The fact that $V_{darea}/F$ was very high in rainbow trout is due to low bioavailability. In this study, $V_{darea}/F$ decreased due to increase in body size. Previous studies have obtained different results regarding $V_d$ change. Due to the size increase in trout, the $V_d$ of trifluralin increased from 2.07 to 3.24 L/kg (*Schultz & Hayton, 1994*), while the $V_d$ of di-2-ethylhexyl phthalate decreased from 0.64 to 0.23 L/kg (*Tarr, Barron & Hayton, 1990*). Lipid, water and protein contents vary depending on the size increase in trout. In addition, as the size increased, the carcass ratio of the dry weight of the fish increased, while the liver ratio decreased (*Weatherley, 1990*). Trifluralin is a lipophilic substance and the increase in $V_d$ has been attributed to an increase in lipid content with body size (*Schultz & Hayton, 1994*). The di-2-ethylhexyl phthalate is a hydrophilic substance and its concentration in liver tissue is approximately 40 times higher than in adipose tissue (*Daniel & Bratt, 1974*). The decrease in $V_d$ of di-2-ethylhexyl phthalate may be due to the shrinkage of the liver in parallel with the increase in body size. In fish, OTC has a special affinity for the liver and

its concentration in this tissue is considerably higher than in plasma (*Corum et al., 2023b*). In this study, the change in $V_{darea}$ depending on size may be due to the physiological difference mentioned above. It was determined that the CL/F of OTC decreased due to body size increase in rainbow trout. Similarly, the CL of trifluralin decreased from 131 to 22.4 mL/h/g depending on body size in trout (*Schultz & Hayton, 1994*). Metabolic activity varies depending on body size (*Weatherley, 1990*; *Schultz & Hayton, 1994*). In addition, due to the increase in body size, the relative organ weights of excretion organs such as liver and gills, which play a role in excretion in fish, decreased (*Weatherley, 1990*). Considering that hepatobiliary excretion is important in the elimination of OTC in trout (*Björklund & Bylund, 1990*), changes in liver and metabolic activity depending on body size may be the reason for the decrease in CL/F.

The $C_{max}$ of OTC in the small, medium and large body size groups was $0.66 \pm 0.07$, $0.87 \pm 0.12$ and $1.11 \pm 0.12$ (oral, 60 mg/kg, $8 \pm 0.5$ °C), respectively. The oral $C_{max}$ of OTC at a dose of 50–100 mg/kg in rainbow trout was between 0.35 and 5.8 μg/mL (*Rigos & Smith, 2015*). This variability in $C_{max}$ of OTC may be due to differences in drug formulation, water temperature and body size (*Corum et al., 2023b*). It was observed that $C_{max}$ and AUC values of OTC increased depending on the body size increase in rainbow trout. Since the water temperature and drug formulation used in the groups were the same, the possible reason for this situation may be physiological differences depending on body size. Oral absorption of OTC is influenced by variables including the width and blood supply of the absorption area, the degree of ionization, and gastric emptying time. OTC is digested from the gastrointestinal tract by 60% in humans and 7–9% in rainbow trout (*Cravedi, Choubert & Delous, 1987*). This difference in OTC digestion is due to the pH difference in the duodenum in humans and trout (*Corum et al., 2023b*). The pH of the duodenum varies depending on the size of the fish (*Matthee et al., 2023*), which affects the degree of ionization of OTC and therefore its absorption. Additionally, physiological differences are observed in the digestive system of fish at different life stages (*Matthee et al., 2023*). The $C_{max}$ and AUC are formed by absorption extent, CL, and $V_d$ of drug. The increase in $C_{max}$ from small to large size may be due to changes in absorption extent, CL/F, and $V_{darea}/F$ depending on fish size.

Pharmacokinetic/pharmacodynamic modeling is used the establish the appropriate dosage regimen of antibiotics. Pharmacokinetic/pharmacodynamic data, including $C_{max}$/Minimum inhibitory concentration (MIC), AUC/MIC, and T>MIC, are utilized to assess the clinical effectiveness of OTC. However, it is not known what these values should be for OTC treatment to be effective (*Rigos & Smith, 2015*; *Toutain et al., 2021*). To determine appropriate pharmacokinetic/pharmacodynamic values, it is important to know the MIC and plasma protein binding ratio. However, the fact that the *in vitro* and *in vivo* MIC values are different and the binding to plasma proteins is atypical and nonlinear makes it difficult to determine the appropriate pharmacokinetic/pharmacodynamic data for tetracyclines (*Toutain et al., 2021*). Therefore, we evaluated the antibacterial effect by taking into account the time when the OTC concentration remained above the MIC value. The MIC value of OTC for susceptible bacteria isolated from fish has been reported to be 0.125−0.75 μg/mL and the susceptible breakpoint to be ≤1 μg/mL (*Rigos & Smith,*

*2015*; *CLSI, 2020*). In this study, the plasma concentration of OTC reached 1 μg/mL only in the large size group, while it reached 0.87 and 0.66 μg/mL in the medium and small size groups, respectively. However, OTC is used repeatedly in bacterial fish diseases. When OTC was administered repeatedly to rainbow trout at a dose of 60 mg/kg for 7 days, $C_{max}$ increased from 1.60 to 7.82 μg/mL (*Corum et al., 2023b*). These results show that OTC accumulates significantly in the body after repeated administration. Considering that OTC was administered as a single dose in this study, it can be thought that a susceptible breakpoint ($\leq$1 μg/mL) can be reached after multiple administration in small and medium sizes.

There are significant limitations to this study that may impact its evaluation. In order to reduce labor force requirements and facilitate application, drugs are commonly administered to fish in the form of medicated feed (*Terzi et al., 2020*; *Uney et al., 2021*). However, medicated feed leach in water (50%) and feed intake loss (25%) during the diseased condition of fish (*Amit et al., 2017*). In contrast, we administered OTC *via* oral gavage in this study to ensure a precise dose and prevent potential drug loss due to feeding. This study was carried out on healthy fish but the pharmacokinetics of OTC changes in disease (*Uno, 1996*). In order to reduce the likelihood of stress and trauma in the fish, both drug administration and blood collection were conducted under anesthesia. However, anesthesia may change pharmacokinetics because it affects the stress response and physiological processes (*Wagner, Singer & Scott McKinley, 2003*). Since serial blood collection is not physiologically possible, especially from small and medium-sized fish, six different fish were used at each sampling time. A single dose OTC was administered to rainbow trout. However, OTC is used repeatedly in bacterial disease and repeated administration alters its pharmacokinetics (*Corum et al., 2023b*). This research was carried out at 8 ± 0.5 °C water temperature. However, the water temperature for trout farming varies by 20 °C. In fish, which are poikilotherm creatures, the pharmacokinetics of OTC (*Björklund & Bylund, 1990*), and MIC values of drugs change depending on water temperature (*Terzi et al., 2020*), so the data obtained in this study may not be suitable for all trout farming conditions.

## CONCLUSIONS

The pharmacokinetics of OTC varied significantly depending on body size. Due to the increase in body size, the CL/F of OTC decreased while its plasma concentration increased. After a single dose of 60 mg/kg, the plasma concentration of OTC reached the susceptible breakpoint concentration only in the large body size group. However, OTC is used repeatedly in bacterial infections and shows significant accumulation in the body after repeated use. Therefore, there is a need to conduct pharmacokinetic and pharmacodynamic studies of OTC in fish of different body size and residue before human consumption.

### Abbreviations

| | |
|---|---|
| **AUC** | Area under the plasma concentration *versus* time curve |
| **CL/F** | Total body clearance |
| **$C_{max}$** | Peak plasma concentration |

| HPLC | High-performance liquid chromatography |
| OTC | Oxytetracycline |
| $t_{1/2\lambda z}$ | Terminal elimination half-life |
| $T_{max}$ | Time to reach peak plasma concentration |
| $V_{darea}/F$ | Volume of distribution |

## ACKNOWLEDGEMENTS

We would like to thank Assoc. Prof. Dr. Ibrahim Ozan Tekeli, who lost his life in the earthquake that occurred in Hatay/Türkiye on February 6, 2023, for his contributions in the experimental phase of the study. We remember him and his daughters Nefes and Eylul with respect.

### Funding

This study was supported by The Coordination of Scientific Research Projects, University of Hatay Mustafa Kemal, Türkiye (Project No. 22.GAP.007). The funders had no role in study design, data collection and analysis, decision to publish, or preparation of the manuscript.

### Grant Disclosures

The following grant information was disclosed by the authors:
The Coordination of Scientific Research Projects, University of Hatay Mustafa Kemal, Türkiye: 22.GAP.007.

### Competing Interests

Önder Yıldırım is an Academic Editor for PeerJ.

### Author Contributions

- Orhan Corum conceived and designed the experiments, performed the experiments, analyzed the data, prepared figures and/or tables, authored or reviewed drafts of the article, and approved the final draft.
- Erdinc Turk performed the experiments, authored or reviewed drafts of the article, and approved the final draft.
- Duygu Durna Corum performed the experiments, analyzed the data, prepared figures and/or tables, authored or reviewed drafts of the article, and approved the final draft.
- Ertugrul Terzi performed the experiments, authored or reviewed drafts of the article, and approved the final draft.
- Mustafa Cellat performed the experiments, authored or reviewed drafts of the article, and approved the final draft.
- Önder Yıldırım conceived and designed the experiments, prepared figures and/or tables, authored or reviewed drafts of the article, and approved the final draft.
- Kamil Uney conceived and designed the experiments, analyzed the data, prepared figures and/or tables, authored or reviewed drafts of the article, and approved the final draft.
## Animal Ethics

The following information was supplied relating to ethical approvals (*i.e.*, approving body and any reference numbers):

The experimental was approved (2021/23) by the Kastamonu University Animal Experiments Local Ethics Committee (Kastamonu, Türkiye).

## Data Availability

The raw data are available in the Supplemental Files.

## Supplemental Information

Supplemental information for this article can be found online at http://dx.doi.org/10.7717/peerj.17973#supplemental-information.

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
