# Peer review of "Effect of body size on the oral pharmacokinetics of oxytetracycline in rainbow trout (Oncorhynchus mykiss)"

_PeerJ, doi:10.7717/peerj.17973_

## Round 0.1 · original submission · Minor Revisions

Dear Dr. Yildirim,

Thanks for sending us your manuscript. I have chosen to accept it with minor revisions. Please consider making the necessary adjustments based on the 1 and 2 reviewers' suggestions and corrections.

Sincerely,

Reviewer 1 ·

Basic reporting

The manuscript is objective, well written and in accordance with the basic reporting standards.

Experimental design

The research is original, the research question is well defined and relevant.
Methods are described with sufficient detail and information to replicate.

Validity of the findings

Conclusions are well stated, linked to original research question and limited to supporting results.
It was very important that authors showed the study limitations in the last paragraph of discussion.

Additional comments

Line 140 – Authors must have meant “- 80 °C”.
Line 234 – I suggest to substitute “depending in body size” by “depending on body size”.
I recommend that the abbreviations are presented unabbreviated the first time they appear. It should be considered a list of abbreviations.

Reviewer 2 ·

Basic reporting

In the submitted manuscript, the authors examined the oxytetracycline effect based on the body size of trout, so the topic is worth investigating. However, the selection of plasma by the authors is appropriate for the pharmacokinetic and pharmacodynamics of oxytetracycline. The authors used different body sizes of fish, which is crucial in developing a proper drug regime.
The experiment reported in this paper was clearly described, While it is an interesting topic in the Aquaculture field, I suggest some corrections and modifications that would improve the manuscript before accepting its publication.
It would be nice to justify or explain the toxicity range of OTC in the line number 91. Since it is not clear author used a USFDA dose or a new dose selected for the dose regime.

Experimental design

Line-122 to 123 exposed to?
Additionally, six different fish were used at each sampling time to minimize the stress and pain the fish would be exposed to.
Line – 136
How is antibiotic dose determined for trout? Is any previous toxicity test conducted?

Validity of the findings

Authors mentioned MIC in paragraphs 257 to 271, it may differ from normal feed premix to direct gavage and immersion method. Line number 279 mentioned that medicated feed is preferred. Fish in the current work was mainly focused on plasma and drug optimisation. Authors can mention that the medicated feed regime may vary due to the leaching effect of OTC in water and during disease fish may not feed or be sluggish in behaviour.
Authors can add a sentence as per Selvin 2010 in line number 279 “However, medicated feed leach in water (50 %) and feed intake loss (25 %) during the diseased condition of fish”
fish”
In conclusion
Line 301
Authors can also mention edible parts of fish such as “different body size and residue before human consumption”.

Additional comments

There are some references missing
Line- 61
Line- 79.

Reviewer 3 ·

Basic reporting

No comment

Experimental design

No comment

Validity of the findings

No comment

Additional comments

See suggested edits in the attached PDF

Annotated reviews are not available for download in order to protect the identity of reviewers who chose to remain anonymous.

---

## Round 0.2 · accepted · Accept

Dear Author,
After reviewing your submissions and corrections, I consider your manuscript to be accepted for publication in accordance with PeerJ's procedures.